# Line Shape Analysis and Dynamic Response of Ballastless Track during Jacking Rectification Fixing

**DOI:** 10.3390/ma15228265

**Published:** 2022-11-21

**Authors:** Wei Chen, Chao Wang, Linhong Fang, Chao Liu, Zhiping Zeng, Ping Lou, Tianqi Zhang

**Affiliations:** 1School of Civil Engineering, Central South University, Changsha 410075, China; 2National Engineering Research Center of High-speed Railway Construction Technology, Central South University, Changsha 410075, China; 3School of Civil Engineering, Guangzhou University, Guangzhou 510006, China

**Keywords:** high-speed railway, CRTS II slab ballastless track, jacking rectification fixing, dynamic response

## Abstract

In order to study the railway line deformation and dynamic response of ballastless track structure under train load during jacking rectification fixing, a three-dimensional numerical model of the CRTS II slab ballastless track on subgrade is established by using the finite element method. The line deformation rule and local damage rule of ballastless track under jacking force are analyzed. The dynamic response laws of track structure and subgrade bed are compared considering four different connection modes between the base plate and subgrade bed under different train speeds in the process of jacking rectification fixing. The results show that jacking force and dissociation length have a small influence on the deviation value and the critical jacking force should be smaller than 375 kN in single point jacking. Under the condition of multi-point jacking, when the jacking loading length equals to 5 slabs, the critical jacking force should be smaller than 275 kN and the maximum lateral deviation value is about 22.11 mm. It is necessary to restrict the speed of passing trains to no more than 150 km/h during the jacking rectification fixing for dissociation condition without temporary restraint. When temporary restraint is applied, the speed of the train can be increased appropriately according to the actual situation. The above study results could be used as a theoretical reference for the ballastless track deviation correction.

## 1. Introduction

CRTS II slab ballastless track with high stability is one of the main forms of track structure used in China’s high-speed railways, which is widely used in high-speed railway main lines at home and abroad [1,2]. In the railway operation process, track alignment and line shape offset occurred due to complex geology, uneven foundation settlement and other factors caused by high-speed train, which may affect the operation and safety of traffic. Railway line offset in a small range can be rectified by adjusting the fasteners [3,4]. Due to a variety of reasons, the lateral displacement of the railway line during operation is greater than the allowable value of operation, and the longitudinal serpentine shape of the railway line or a long waveform that affects the safety of normal operation will appear. This paper intercepts a section of the line shape change of the railway line and draws a schematic diagram, as shown in Figure 1a. The black dotted line in the picture represents the standard position of the railway line, and the red solid line in the picture indicates that the railway alignment has a large lateral displacement deviation after a long period of operation. Figure 1b shows an on-site view of the use of a jacking rectification fixing to push the offset track slab to the previously designed position during the skylight period of train operation. After that, the line linearity should be restored and the smoothness must be ensured during train driving [4].

The rectification technology of the CRTS II slab ballastless track in the subgrade section is divided into two types [4,5,6]: mechanical rectification and grouting rectification. Mechanical rectification is mainly lifting the track by jack and then using the horizontal jack for lateral rectification of track slab, and finally fill polymer grouting material in the lifting gap; the implementation of grouting rectification technology relies on the reasonable arrangement of grouting lifting holes and grouting holes, releasing the constraint (dissociation) between the base plate and subgrade bed through temporary debonding agent, and then through lateral jacking to realize the rectification of track structure, as shown in Figure 1c. In order to ensure the safe operation of the train during the separation operation, temporary constraints are imposed between the base plate and the subgrade bed, as shown in Figure 1d. In Figure 1d, the temporary restraint is a temporary fixation of the track slab to the subgrade bed, restricts the horizontal displacement of the track slab that has been separated, and ensures the smoothness of the train operation line. After rectification, the dissociated interface between the base plate and the subgrade bed will be restored by grouting material.

During the normal operation of high-speed railways, the safety of trains is closely related to the dynamic characteristics of the track structure [7,8]. The dynamic performance indexes [9,10] of the track structure is instructive to the safe train running conditions. Generally, it takes some time to carry out the jacking rectification fixing on the existing railway line whose geometry offset is too large, which can’t be completed in one skylight time. However, the existing railway line still needs to have trains running in the daytime without interrupting transportation. In order to ensure the safe passage of trains, it is important to explore the dynamic response [11,12,13] of the track structure during the jacking rectification fixing interval.

In order to study the structural dynamic response and structural damage [14,15] caused by the magnitude and range of jacking force in the process of jacking, and to explore the correlation between train speed and structural response during jacking rectification fixing. In this paper, based on the actual construction process, a three-dimensional numerical model of the CRTS II slab ballastless track is established using Abaqus finite element software to analyze the damage of interface [16,17] between mortar layer and the wide-narrow joint and the line shape [18] of the ballastless track under single point jacking and multi-point jacking. Meanwhile, the jacking parameters influence during rectification construction process is studied. Then by writing the subroutine DLOAD to apply the train load [19,20], the dynamic response of the track structure during the jacking rectification fixing interval is analyzed on the basis of the existing numerical model [21]. The vibration acceleration and displacement of the track slab and the subgrade bed under different temporary connection methods between the base plate and the subgrade bed as well as the dynamic response under different train speeds [22] are investigated, and combining with relevant codes, the safe train speed [23] range during the jacking rectification fixing interval is discussed. There are two main innovations in this paper: first, the relationship between jacking force and structural damage is established in single-point jacking and multi-point jacking conditions; second, during the interval of jacking rectification fixing, the structural response of high-speed trains under different working conditions is studied, and on this basis, measures are made to limit the train speed to the allowable value or to add a temporary constraint. Furthermore, it provides a theoretical reference for CRTS Ⅱ slab ballastless track in the process of jacking.

## 2. Numerical Model

Abaqus software is used to establish a CRTS II slab ballastless track model for the subgrade section that matches the actual situation. The model consists of subgrade ballastless track, as shown in Figure 2a. The numerical model from top to bottom is rail, rail bearing platform, track slab, mortar layer, base plate and subgrade. According to references [24,25], the length of the whole model along the track direction is 208 m, a total of 32 plates are long (each plate is 6.5 m), the rail is connected with the rail platform through fasteners, and the track slabs are connected by the wide-narrow joint; the subgrade is trapezoidal, the height is 4.5 m, the top width is 8.6 m, the bottom width is 24.2 m and the slope is 30°, in which the longitudinal distance of the rail bearing platform along the line is 0.65 m. The cross-sectional dimensions are shown in Figure 2b. The overall size of the model is shown in Figure 2c, and the wide-narrow joint are distributed in the model as shown in Figure 2d.

The rail is in the form of CHN60 section [26], with a mass of 60.64 kg·m^−1^, a cross-sectional area of 77.45 cm^2^, a horizontal axis moment of inertia of 3217 cm^4^ and a vertical axis moment of inertia of 524 cm^4^. In this paper, the steel rail, rail bearing platform, track slab, fastener, reinforcement, mortar layer and base plate all adopt linear elastic constitutive model, and the parameters of each part are shown in Table 1. Drucker Prager ideal elasto-plastic constitutive model is adopted for the surface of subgrade bed, bottom of subgrade bed and subgrade body, and the parameters of each part are shown in Table 2.

Except for soil mass, the materials are all modelled using a linear elastic principal structure model in which the fasteners are connected by three-way Cartesian springs [20], allowing full consideration of the longitudinal resistance, transverse stiffness and vertical stiffness of the fasteners, regardless of the possible three-way rotational degrees of freedom of the fasteners. According to the literature [25,27], the longitudinal, transverse and vertical stiffnesses are taken to be 15.12, 50 and 35 kN/mm respectively; the maximum longitudinal resistance provided by each group of fasteners is 35 kN, the fasteners are arranged at equal spacing and the distance between fastener nodes is 0.65 m. The longitudinal connection reinforcement is simulated using the B31 beam unit. And the role of reinforcement is to effectively limit the expansion of cracks in the track slab. The mortar layer thickness is relatively small and is simulated using the C3D8I solid unit, the rest structures are simulated using the C3D8R solid unit.

During the jacking rectification fixing of the ballastless track, jacking force is applied by jacking, one jack is arranged every 2 m, and a 180 mm × 180 mm steel plate is arranged between jacking and base plate layer to prevent stress concentration. In order to simplify the model, the applied jacking force is regarded as the uniform pressure acting on the base plate, and the area of action is the area of the steel plate. Dissociation, as the name implies, is to release the constraint between the base plate and the subgrade bed surface by mechanical lifting or grouting; in the ABAQUS finite element method, the dissociation between the base plate and the subgrade bed surface is realized by changing the contact model (from the cohesion model to the Mohr-Coulomb model). The cohesion model [28,29] is used to simulate the interaction between the track slab and the mortar layer, the interactions between the wide-narrow joint and the surrounding track slab, and the interaction between the base plate and the subgrade bed without dissociation. By analyzing the damage state of the cohesive model, it is possible to determine whether there is a detaching between the layers of the track structure. The normal and tangential-displacement relationships of the cohesion model are described by the bilinear tension-displacement rule in the model. The cohesion model parameters are based on literature [25,30], with a normal cohesion strength of 1.792 MPa, an interface stiffness of 708.485 MPa/mm, a critical fracture energy of 0.0252 MJ/mm^2^, and with a tangential cohesion strength of 0.956 MPa, an interface stiffness of 63.039 MPa/mm, a critical fracture energy of 0.0252 MJ/mm^2^.

For dissociation, the contact between the base plate and the subgrade bed is described by classical friction theory-Coulomb friction theory (because of the dissociation, there is no cohesive strength between the base plate and the subgrade bed), Ignoring the influence of other factors on friction, only the maximum allowable friction (shear) stress on the contact interface is associated with the contact pressure between the contact bodies [31,32], and the relationship is expressed as follows:(1)τ=μP
where τ is the shear stress, μ is the friction coefficient, which is taken as 3.3 according to the reference [27] and P is the contact pressure on the contact surface. The longitudinal reinforcement is embedded in the track slab and the wide-narrow joint using the “Embedded” method. The contact between the mortar layer and the base plate is made using the “Tie” connection.

## 3. Mechanical Behaviour under Jacking Pressure

### 3.1. Single Point Jacking

Track alignment refers to the spatial position of the railway centerline, consisting of straight lines and curves on the plane and longitudinal section. When the deviation of the alignment causes the track geometry to be offset beyond the permissible adjustment range of the fasteners, the alignment can be restored to smoothness by jacking rectification fixing. The deviation of the alignment of the track structure in mechanical lateral jacking rectification fixing is related to the dissociation length of the base plate bottom and the value of jacking force during the single point jacking [27]. The dissociation length and jacking force are chosen as the study parameters, and the dissociation length is respectively equal to the length of 1, 3, 5, 7, 9, 11 and 13 track slabs. Single point jacking forces of 400, 800, 1200, 1600 and 2000 kN are respectively applied at the base plate in the middle of the model. A schematic representation of the single point jacking rectification and the corresponding position of the measuring line are shown in Figure 3.

The results of the CRTS II slab ballastless track model for the subgrade section established in the literature [27] are selected for validation, and the maximum lateral deviation of the corresponding track slab at the single point jacking position is calculated as shown in Figure 4.

Figure 4 shows the effect of the dissociation length on the amount of jacking rectification displacement under single point jacking. With the increase of the dissociation length and jacking, the maximum correction displacement of the track structure gradually increases, and the minimum correction displacement of the jacking is 0.228 mm when the dissociation length is 0 (no dissociation). However, the effect of the dissociation length on the amount of corrective displacement is limited. When the jacking force is 2000 KN and the dissociation length is 5 plates, the maximum corrective displacement is 2.56 mm, and when the dissociation length continues to increase to 13 plates long, the maximum corrective displacement is only 2.67 mm, and the growth of corrective displacement is only 0.11 mm. The above calculation results are compared with those of the literature [27], and the deviation of the maximum corrective displacement of the track structure for different dissociation cases is within 0.2 mm. At the same time, they have the same curve variation trend, which verifies the correctness of the finite element model established in this paper. Based on this result, mechanical lateral jacking rectification fixing in practical engineering should be prioritized for the base plate bottom dissociation.

The overall alignment of the track is investigated considering the variations of the dissociation length of the base plate bottom and jacking force. The lateral deformation of track slab along the longitudinal direction of railway line for a single point jacking force of 2000 kN with different dissociation lengths (The position of the measuring line, the same as below.) are shown in Figure 5. The difference between the lateral deviation of the track for the dissociation lengths of 1, 3 and 5 slabs is noticeable. When the length of dissociation reaches 5 slabs and the length of dissociation continues to increase, the deviation no longer increases significantly. The track deviation curve has a groove shape and has a maximum at the position of the jacking point. It can be concluded that the single point jacking has a limited range of deviation correction (railway line shape restoring) and is suitable for small scale deviation correction at specific locations in practical engineering.

The lateral deviation curve of the track is shown in Figure 6 for a dissociation length of 5 slabs with different jacking forces. It can be seen that the increase of single point jacking force has a facilitating effect on the increase of lateral deviation, for every 400 kN increase of jacking force, the lateral deviation of the track increases by nearly 0.5 mm. Therefore, in the process of deviation correction of the railway line shape, when the deformation stress of the structure under the jacking force does not exceed the allowable value of the material stress, the jacking force can be increased moderately to achieve the increase of lateral deviation.

### 3.2. Multi-Point Jacking

From the previous analysis, it is known that the deviation correction of the railway line shape by single point jacking is limited, and the maximum deviation correction is within 3 mm under the jacking force of 2000 kN (Figure 6); in the actual correction and fixing process, the effect of single point jacking is not so obvious from the perspective of the maximum correction amount alone, and its action range is limited. Therefore, further analysis of multi-point jacking for the deviation correction are carried out (Figure 7). Every 2 m a jack is arranged and a 180 mm × 180 mm steel plate between the jack and the base plate is arranged. The jacking force in the model is represented by the uniform distribution of force which is applied on the side of the base plate, as shown in Figure 7.

For multi-point jacking, a comparative analysis for the influence of dissociation is also carried out. The dissociation length is set to equal to the length of base plates which are pushed by jacks. The maximum lateral deviation of the track with and without dissociation at a jacking force of 400 kN is shown in Figure 8. The effect of dissociation on the deviation of the track during jacking rectification is significant; the maximum deviation is only 18.2 mm for a multi-point jacking with 400 kN jacking force without dissociation; and the maximum deviation increases with the increase of jacking loading length (Figure 7a) in the dissociation case, with a positive correlation between the jacking loading length and the maximum deviation of the track structure. With a jacking force of 400 kN and a jacking loading length of 13 slabs, the maximum deviation of the track structure can reach 67 mm, that means if the alignment (line shape) of the existing railway line needs to be corrected, a maximum of 67 mm in the lateral direction can be adjusted by this method under the above condition.

When the jacking loading length remains the same (jacking loading length of 5 slabs), the jacking force increases from 100 kN to 500 kN and the lateral deviation of the track structure also increases from about 7.86 mm to 40.39 mm, as shown in Figure 9. Therefore, the desired deviation can be achieved by adjusting the value of the jacking force (Figure 9) and the value of the jacking loading length (Figure 8).

The lateral deviation curves of the track for multi-point jacking with a loading length of 5 slabs are shown in Figure 10. As can be seen from Figure 10 in the case of dissociation and loading length of 5 slabs, the overall deviation curve of the track shows a symmetrical “groove” shape, and the deviation reaches the maximum at the center of jacking. With the increase of jacking force, the deviation also increases, and the increase of deviation at the center jack is about 8 mm/100 kN. After data fitting, the relationship between the deviation and the jacking force can be described as follows:(2)y=−0.4053+0.08187x

### 3.3. Damage Analysis for Wide-Narrow Joint during Single Point Jacking

According to the damage mechanism of the bilinear cohesive connection model, it can be seen that the relative displacement between the mortar layer and the wide-narrow joint will occur in the process of jacking rectification fixing. With the increase of relative displacement, when the tension value reaches the bearing limit, cracks appear on the interface and continue to develop; When the stress drops to zero, the crack fully expands and the interface bond fails. In this paper, the contact stiffness (CSDMG) of the interface between the above is used to evaluate the bond damage in the process of jacking rectification fixing, and the bond damage of the interface between the mortar layer and the wide-narrow joint is analyzed.

For the single point jacking rectification, the study shows that the contact between the track slab and wide-narrow joint is essentially the contact between the new and old concrete [23]. The interface between the mortar layer and the wide-narrow joint is more prone to damage (Figure 11), so this paper focuses on the interface damage between the mortar layer and the wide-narrow joint within the action of the jacking force. In this paper, the bond damage between mortar layer and the wide-narrow joint is analyzed by evaluating the bond damage in jacking rectification by the contact stiffness (CSDMG) of bond surface. The stiffness drop rate ranges from 0 to 1, when CSDMG = 1, it means that the bonding damage occurs in the interface (occurring crack), when CSDMG = 0, it means that the bonding damage does not occur. The bond performance between the mortar layer and the wide-narrow joint under a single point jacking force of 500 kN is shown in Figure 11.

The relative positions of the wide-narrow joint and the mortar layer in the model are shown in Figure 11a. The meanings of the symbols for plate 0#, plate 1# are the same as in Figure 3. Figure 11b,c show the results of the surface of the wide-narrow joint and the mortar layer off the joint under the action of the single point jacking force, respectively. As shown in Figure 11, during the single point jacking process, the bonding between the mortar layer and the wide-narrow joint in the model area is damaged, and only a local reduction in the contact stiffness of the interface occurs in the slab where the jacking force is applied below, and a separate area with a width of 210 mm and a length of 510 mm appears between the mortar layer and the wide-narrow joint. The separation area is about 19% of the total area of the interface, which indicates that the interface bond between the mortar layer and the wide-narrow joint is slightly damaged during the single point jacking process with 500 kN force.

In order to study the value of the jacking force at the bonding condition of the mortar layer and the wide-narrow joint interface in the single point jacking process, a series of working conditions are simulated. The damage at the surface position on the right side of the jacking center is obtained when different jacking forces are applied at the base plate below the center of 0# (Figure 3) as shown in Figure 12. It can be concluded that when the jacking force is located on the base plate below the center of 0 # slab, the critical jacking force for avoiding the cracking of the interface between the mortar layer and the wide-narrow joint is about in the range of 350 Kn–375 kN. Therefore, we should set the jacking force smaller than 350 kN in this condition during single point jacking rectification to prevent damage of the interface inside CRTS II slab ballastless track structure.

A series of numerical simulations are carried out to study the influence of dissociation length on the critical jacking force when the interface damage just beginning during single point jacking condition. The relation of critical jacking force corresponding to the detachment length of 1 slab, 3 slabs, 5 slabs, 7 slabs, 9 slabs, 11 slabs and 13 slabs is obtained (Figure 13) when the single point jack point is located at the center of 0#. When the dissociation length is 1 slab, the critical jacking force is the largest. With the increasing dissociation length, the critical jacking force presents a downward trend, and when the dissociation length is within the range of 5 slabs, the decline is larger. After more than 5 slabs, the decrease is small, and finally flattens out. That is, when the dissociation length is greater than or equal to 11 slabs, the critical jacking force for avoiding damage is stable at about 375 kN. On this basis, it can be inferred that when the dissociation length is 11 slabs and above, the critical jacking force for avoiding damage of the interface between the mortar layer and the wide-narrow joint can be considered consistent with that when the dissociation length is 11 slabs.

### 3.4. Damage Analysis for Wide-Narrow Joint during Multi-Point Jacking

Based on the single-point jacking damage analysis method, the variation rule of critical jacking force under multi-point jacking condition is studied. In order to exclude the dissociation length effect on critical jacking force, the jacking loading length is set to 5 slabs and the dissociation length in the model is equal to 32 slabs. The simulation results are shown in Figure 14. It can be seen from Figure 14 that the critical jacking force for avoiding damage of the interface between the mortar layer and the wide-narrow joint is about 275 kN under this condition. When the jacking force is less than 275 kN, the integrity of the interface between the mortar layer and the wide-narrow joint is good, the interface is not damaged, and the structure is within the safe range. When the jacking force is greater than 275 kN, a certain damage area appears on the interface, and the damage area increases suddenly with the continuous increase of the jacking force. When the jacking force reaches 500 kN, the percentage of damage area on the interface between the mortar layer and the wide-narrow joint exceeds 50%.

In order to analyze the influence of the dissociation length change on the critical jacking force, the corresponding critical jacking force is calculated under the condition of jacking loading length of 5 slabs when the dissociation length is 5 slabs, 7 slabs, 9 slabs, 11 slabs, 13 slabs, 15 slabs, 17 slabs and 19 slabs. The relationship between different dissociation lengths and the critical jacking force is shown in Figure 15. It can be seen from Figure 15 that the critical jacking force for avoiding damage of the interface between the mortar layer and the wide-narrow joint presents a nonlinear downward trend with the increase of the dissociation length. When the dissociation length is 17 slabs or above, the critical jacking force tends to a stable value, about 275 kN.

## 4. Dynamic Response of the Track Structure during the Jacking Rectification Interval

### 4.1. Train Loading Profile

In this paper, the operation of a train on a track is simulated by applying a moving load [19,33]. In the model, the drop subroutine of ABAQUS software is selected to realize the train load, the implicit integration algorithm is used in the analysis step, and the type of analysis step is dynamic, implicit. The train load is composed of a series of axle weight loads. Assuming a total of M carriages, four-wheel pairs inside a carriage and a train speed of c, the continuous axle weight load [33] generated by the train is shown schematically in Figure 16.

The continuous axle weight load generated by the train is shown in Equation (3) [33]:(3)$$P_{M}=\sum_{n=1}^{M}f_{n}(x-ct)$$

The right part in the equation represents the vertical load on the rail generated by the axle weight of the nth carriage as it moves, as expressed in Equation (4) [33]:(4)$$\begin{array}{l} f_{n}(x-ct)=P_{n1}\delta (x-ct+\sum_{s=0}^{n-1}L_{s}+L_{0})+\\ P_{n1}\delta (x-ct+a_{n}+\sum_{s=0}^{n-1}L_{s}+L_{0})+\\ P_{n2}\delta (x-ct+a_{n}+b_{n}+\sum_{s=0}^{n-1}L_{s}+L_{0})+\\ P_{n2}\delta (x-ct+2a_{n}+b_{n}+\sum_{s=0}^{n-1}L_{s}+L_{0}) \end{array}$$
where $p_{n1}$ and $p_{n2}$ denote the axle weights of the front and rear wheels of the carriage respectively, $L_{n}$ is the length of the carriage, $L_{0}$ is the distance from a set measurement point in the first carriage and $a_{n}$, $b_{n}$ are the distances between axles.

In this paper, CRH380 trains are selected as the basis for the parameters of the DLOAD subroutine, where the high-speed train parameters of CRH380 type [34] are shown in Table 3 below.

### 4.2. Location of Measurement Points

Monitoring points are set up at different locations in the model and the arrangement of each monitoring point is shown in Figure 17. Point A is the rail, point B is the surface of the track slab, point C is the surface of the base plate, point D is the surface of the subgrade bed.

### 4.3. Model Validation

In order to verify the accuracy of the model, the vibration acceleration of the rail and the surface of the base plate under normal operating conditions at a train speed of 300 km/h without dissociation is first calculated, and the time course curves of the vibration acceleration at point A (on the rail) of the rail and point C (on the track slab) of the base plate are shown in Figure 18. The calculation result of point A and C shows a vibration peak at the axle load passing through, and the maximum acceleration fluctuation range in point A is from −600 m/s^2^ to 600 m/s^2^ and the maximum acceleration fluctuation range in point C is from −3 m/s^2^ to 4 m/s^2^. Whether at point A or C, the calculated vibration peak frequency is consistent with the measured curve in literature [34], and the peak value fluctuates slightly with the measured value in literature, but most of the fluctuation range errors of both are within 10%, so it can be considered that the numerical model established in this paper is effective.

### 4.4. Dynamic Response of Track Structures under Different Operating Conditions

In the actual project, as it takes a certain amount of time to complete the rectification, it is very likely that the whole rectification operation cannot be completed within a single skylight time, so temporary restraints need to be applied to the track structure to ensure the normal passage of trains during the whole rectification period. The “Beam-MPC” method is used in the model to apply the temporary restraint, which is shown in Figure 19.

According to the high-speed railway design code [35] and relevant regulations on transport safety management, the dynamic response of the track structure during the jacking rectification fixing interval is investigated for the case of uninterrupted trains running [36,37,38,39]. The vibration acceleration and vibration displacement of the track slab surface under several different connection methods between the base plate and the subgrade bed are analyzed by four different numerical working conditions, namely no dissociation, dissociation (complete dissociation), temporary restraint 1 (temporary restraint at both ends, Figure 19a) and temporary restraint 2 (temporary restraint at both ends + middle, Figure 19b). The dynamic response of the central position of the track slab in the middle of the model under the action of a CRH380 train at a speed of 200 km/h is shown in Figure 20.

From Figure 20a–d, it can be known that the maximum value of the acceleration on the surface of the track slab is 105.6 m/s^2^ when the train is running at a certain speed under no-dissociation. The acceleration on the surface of the track slab under the dissociation condition increases significantly compared to the non-dissociation condition, up to a maximum of 308 m/s^2^. Under the dissociation condition, the acceleration on the surface of the track slab with the temporary restraint is significantly lower than that without the temporary restraint, and it can be seen that the magnitude of the acceleration on the surface of the track slab decreases significantly as more temporary restraints applied. The fluctuation range of acceleration on the surface of the track slab for the temporary restraint 1 condition is from 10.5 to 198.4 m/s^2^, and the maximum acceleration reduction on the surface of the track slab is about 36% compared to the simple dissociation condition without temporary restraint; the fluctuation range of acceleration on the surface of the track slab for the temporary restraint 2 condition is from 9.8 to 145.6 m/s^2^, and the maximum acceleration reduction on the surface of the track slab is about 53% compared to the simple dissociation condition without temporary restraint. The effect of temporary restraint 2 is good in terms of acceleration indicators.

From the time-history curves of the vertical displacements in Figure 20e–h, it can be seen that the vertical displacement caused by the dissociation conditions is the largest under the four operating conditions, but the difference is small; The vertical displacements in the dissociated case with temporary restraint are similar to those of the non-dissociation case. The overall vertical displacement fluctuates between 0.05 and 0.08 mm for all four conditions, with the largest values occurring at the passage of the train axle load.

### 4.5. Influence of Travel Speed

Figure 21 shows the variation of peak acceleration and peak displacement at the center of the track slab in the middle of the model and at the corresponding position of the subgrade bed surface at different train speeds for dissociation, non-dissociation, temporary constraint 1 and temporary constraint 2. The difference in peak acceleration for the same train speed between the dissociation and non-dissociation conditions is obvious. The corresponding acceleration and displacement response tend to decrease gradually as the temporary restraint is enhanced, with a significant decreasing effect on acceleration and displacement response after the temporary restraint 2 is applied. The acceleration on the surface of the track slab and the surface of the subgrade bed increases with the increase of train speed. During the jacking rectification fixing interval, it is necessary to discuss train speed limitation in order to ensure the safety of train running. According to the Technical Regulations for Dynamic Acceptance for High-speed Railways Construction TB 10761-2013 [9], the vibration acceleration of ballastless track slabs should be less than 300 m/s^2^ and the reference limitation value of vertical displacement in the slab is 0.2 mm. From Figure 21a it can be seen that the vibration acceleration of the track board exceeds 300 m/s^2^ at a train speed of 200 km/h (the dissociation condition), with temporary constraint 1 applied, the vibration acceleration of the track slab exceeds 300 m/s^2^ at a train speed of 300 km/h, and with temporary constraint 2 applied, the vibration acceleration of the track slab is within the standard limit when the train speed reaches 350 km/h. It can be seen that it is necessary to restrict the speed of the passing trains during the construction interval, which should be limited to no more than 150 km/h (with dissociation); when temporary restraint is applied, the travelling speed can be increased appropriately in conjunction with the actual situation.

As shown in Figure 21c, the acceleration of the subgrade bed surface is relatively large under the dissociation condition, and for the non- dissociation, temporary constraint 1 and temporary constraint 2 conditions, the difference between the acceleration of the subgrade bed surface at a train speed of 350 km/h and 100 km/h is very small, around 0.2 mm/s^2^.

As shown in Figure 21b,d, the kinetic indexes of the track structure under the action of the train with speed in the range of 100–350 km/h and under four different operating conditions are within the vertical displacement limits, and the effect of the change in train speed on the peak vertical displacement of the track slab and the subgrade bed surface is minimal. For the dissociation conditions, the change in train speed has some effect, but the overall difference is within a small range. It can be known that the effect of train speed on the vertical displacement of the subgrade bed surface is very small and the effect on the vertical displacement of the track slab and the subgrade bed surface caused by the change in train speed during the jacking rectification fixing is not significant. As the temporary restraint enhanced, the dynamic response of the structure decreases. The temporary restraint has a good effect, so it is important to use the temporary restraint during the rectification, especially after the dissociation process is completed to ensure the train running safety.

## 5. Conclusions

In the study, based on the CRTS II slab ballastless track jacking rectification model of subgrade section, the structural responses under single point pushing, multi-point pushing and high-speed moving load are studied, and the following main conclusions are obtained.

For the CRTSII slab ballastless track in the single point jacking rectification, the deviation value increases with the increase of jacking force and the dissociation length. When the dissociation length exceeds 5 slabs, the influence of the dissociation length on the deviation is significantly weakened. For avoiding the damage of interface between the mortar layer and the wide-narrow joint, the jacking force should be smaller than 375 kN in some conditions.In the process of multi-point jacking, the deviation value is also related to the jacking force and the dissociation length. When the jacking loading length equals to 5 slabs, The relationship between the deviation and the jacking force can be described as y=−0.4053+0.08187x. For avoiding the damage of interface between the mortar layer and the wide-narrow joint, the jacking force should be smaller than 275 kN in above condition, and the lateral deviation will be smaller than 22.11 mm.The acceleration of the track slab caused by the train in the dissociation condition is large, and has a significant increase with the increase of the train speed. The acceleration of the track slab has a significant decrease with the application and enhancement of the temporary restraint. It is necessary to restrict the speed of passing trains to no more than 150 km/h during the jacking rectification fixing for dissociation condition without temporary restraint.The train speed in the range of 100 km/h–350 km/h has very little effect on the vertical displacement of the track slab and the subgrade bed surface under the four operating conditions.

## Figures and Tables

**Figure 1 materials-15-08265-f001:**
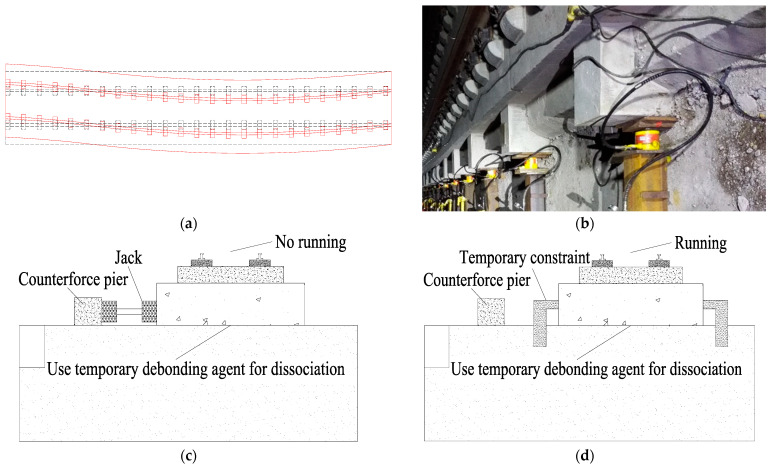
The overview diagram. (**a**) schematic diagram before and after the change in the route shape of the railway line; (**b**) an on-site view of the use of a jacking rectification fixing; (**c**) the cross section schematic diagram of the use of a jacking rectification fixing; (**d**) schematic diagram of the temporary constraint.

**Figure 2 materials-15-08265-f002:**
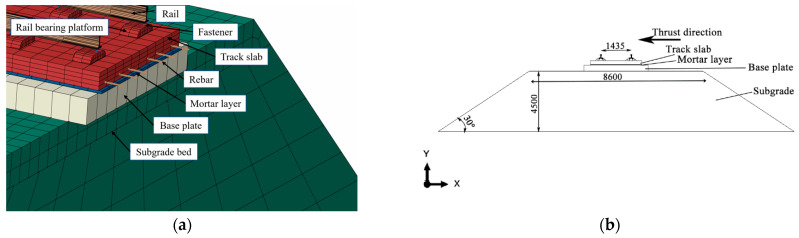
CRTS II slab ballastless track model on subgrade. (**a**) composition of each part of the model, (**b**) model cross section size, (**c**) overview of numerical model, (**d**) model partial enlargement of lateral view.

**Figure 3 materials-15-08265-f003:**
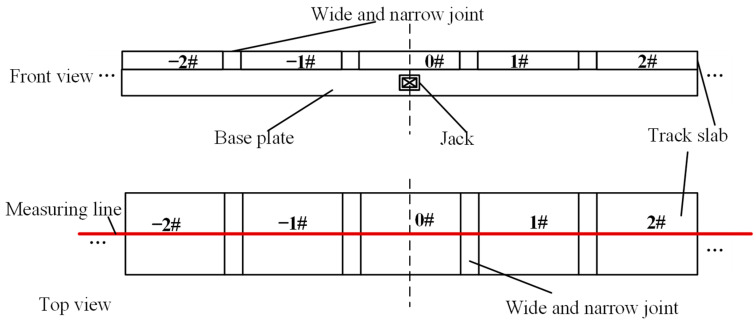
Diagram of single point jacking.

**Figure 4 materials-15-08265-f004:**
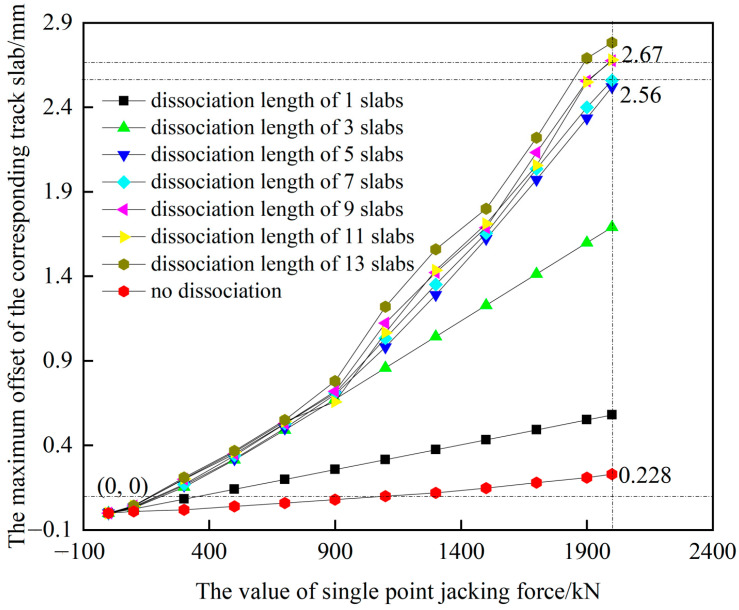
Effect of dissociation length on single point jacking rectification.

**Figure 5 materials-15-08265-f005:**
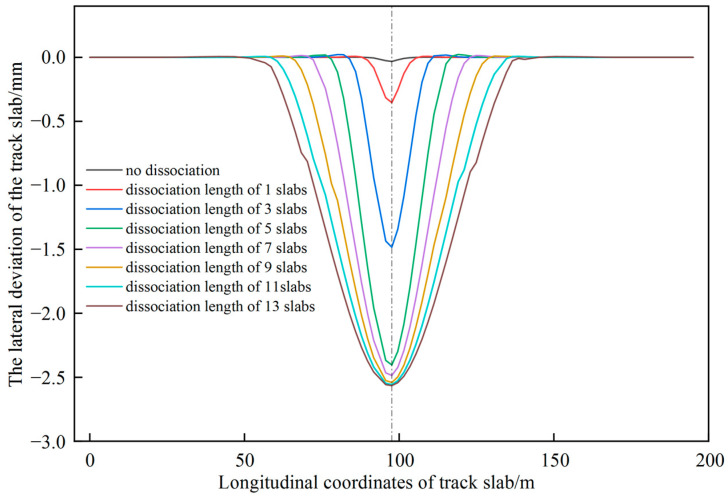
Track deviation curves of different dissociation lengths under single point jacking.

**Figure 6 materials-15-08265-f006:**
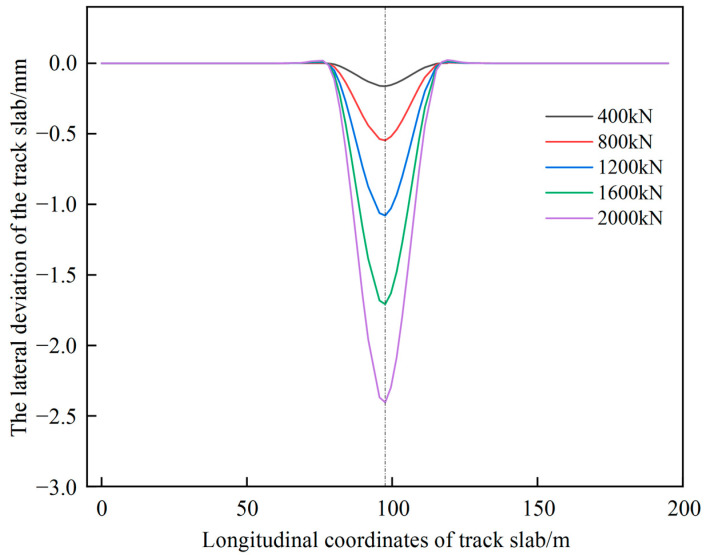
Lateral deviation curves of different jacking forces under single point jacking.

**Figure 7 materials-15-08265-f007:**
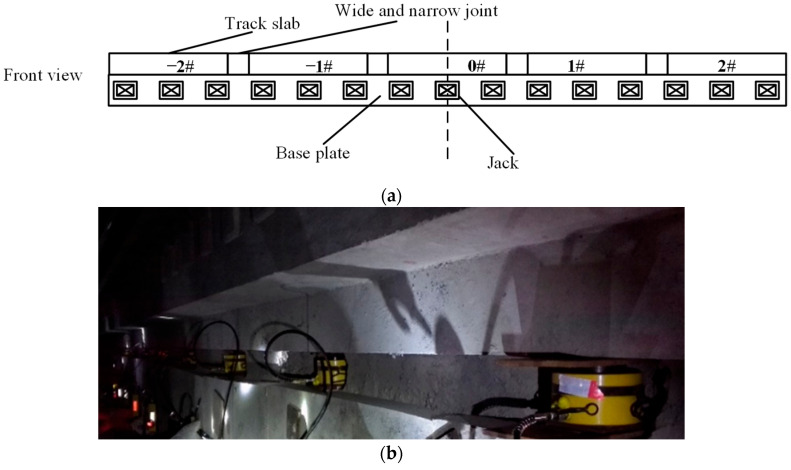
Multi-point jacking, (**a**) the schematic diagram, (**b**) picture on site.

**Figure 8 materials-15-08265-f008:**
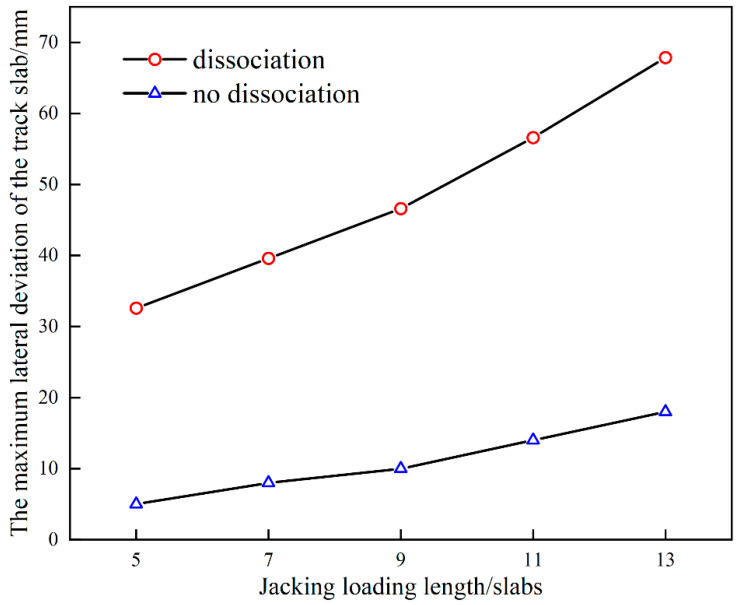
Influence of jacking loading length during multi-point jacking.

**Figure 9 materials-15-08265-f009:**
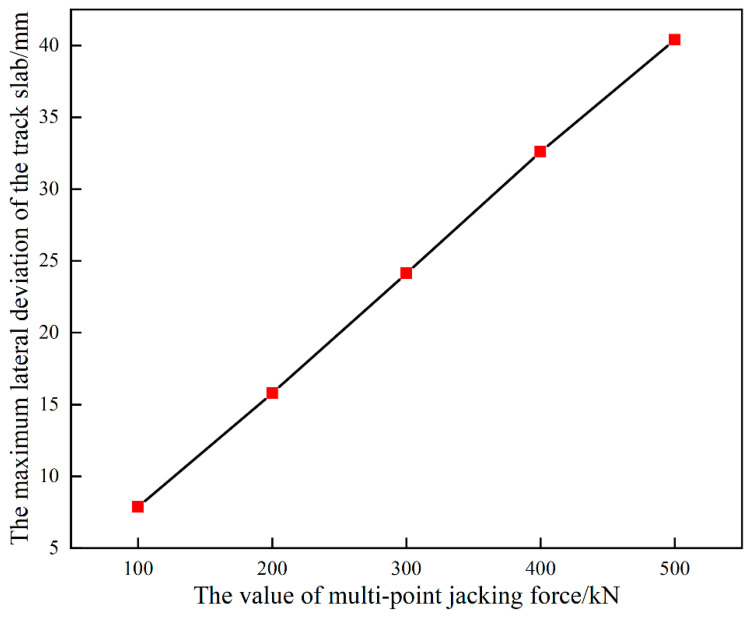
Maximum lateral deviation of track under different jack force.

**Figure 10 materials-15-08265-f010:**
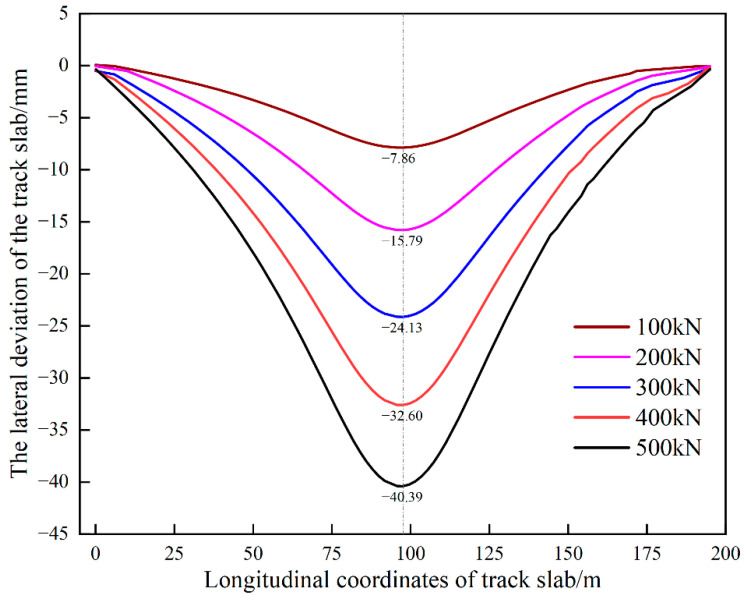
Lateral deviation curves of different jacking forces under multi-point jacking.

**Figure 11 materials-15-08265-f011:**
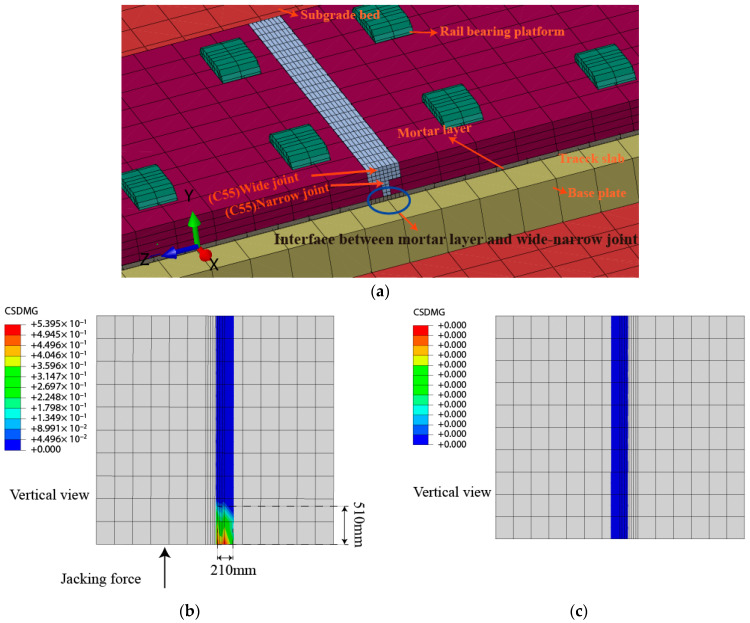
Damage and failure of interface between mortar layer and wide-narrow joint under 500 kN jacking force, (**a**) Schematic diagram of the wide-narrow joint, (**b**) 0 # slab with jack, (**c**) 1# slab with jack (Figure 3).

**Figure 12 materials-15-08265-f012:**
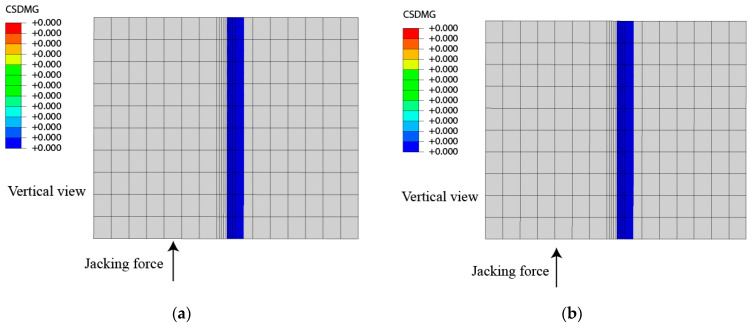
Effect of jacking force on damage and failure of the interface between mortar layer and wide-narrow joint, (**a**) 300 kN jacking force; (**b**) 350 kN jacking force; (**c**) 375 kN jacking force; (**d**) 400 kN jacking force; (**e**) The ratio of the separated area to the total interface area when different jacking forces are applied.

**Figure 13 materials-15-08265-f013:**
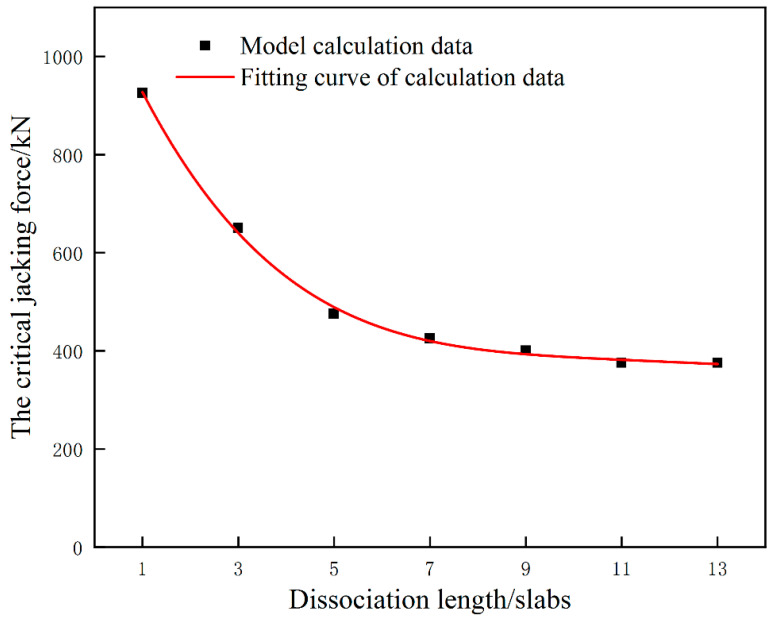
The critical jacking force corresponding to different dissociation lengths under single point jacking condition.

**Figure 14 materials-15-08265-f014:**
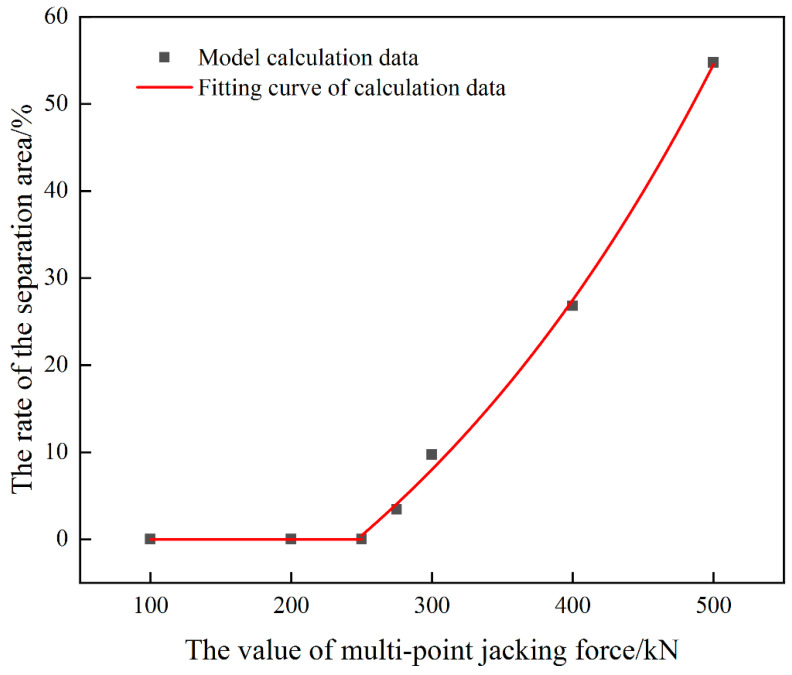
The ratio of the separated area to the total interface area when different jacking forces are applied under the condition of multi-point jacking.

**Figure 15 materials-15-08265-f015:**
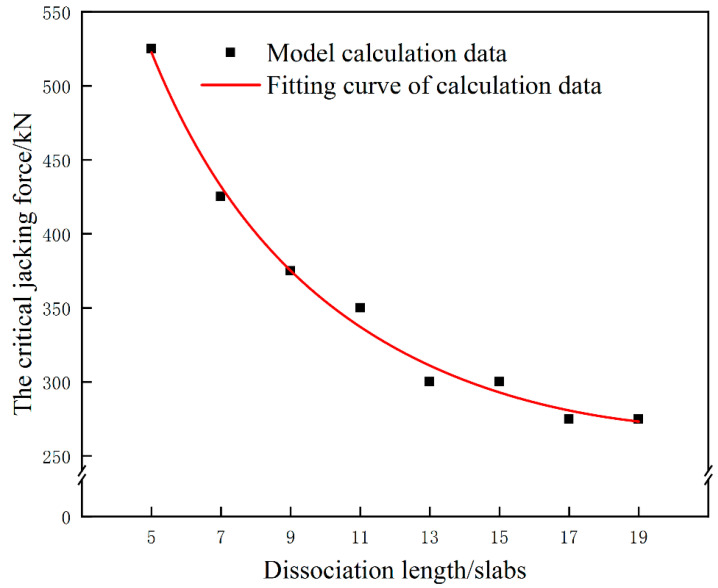
The critical jacking force corresponding to different dissociation lengths under multi-point jacking condition.

**Figure 16 materials-15-08265-f016:**
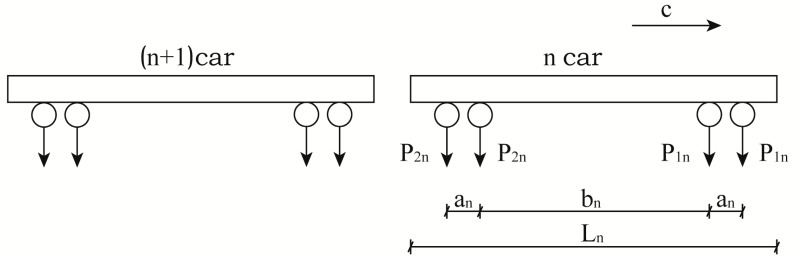
Schematic diagram of train axle load.

**Figure 17 materials-15-08265-f017:**
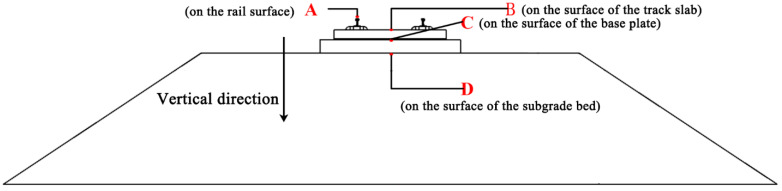
The layout of cross-section measuring points.

**Figure 18 materials-15-08265-f018:**
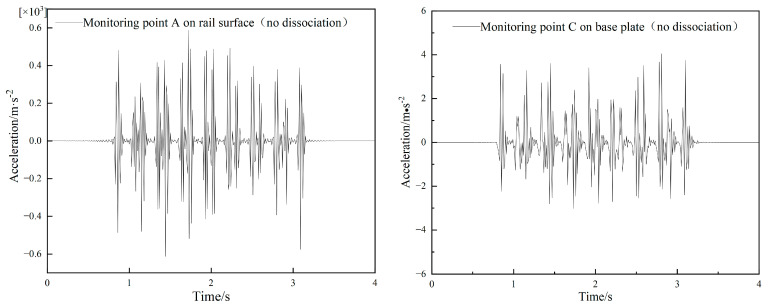
Calculation results of vibration acceleration under normal conditions. (**a**) acceleration time history of measuring point A, and (**b**) measuring point C.

**Figure 19 materials-15-08265-f019:**
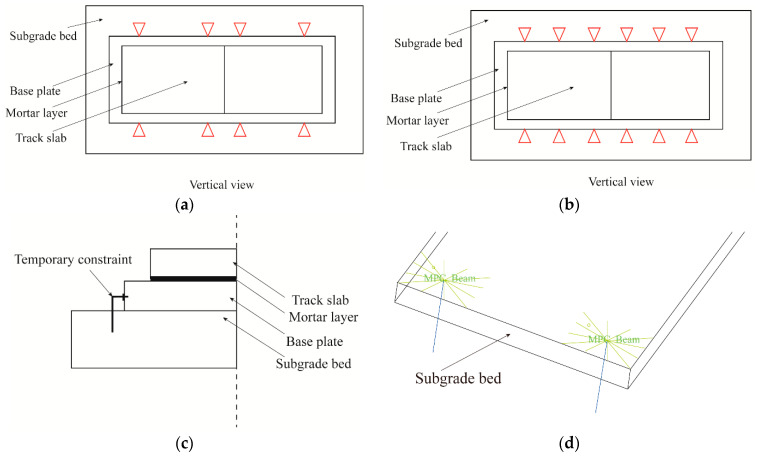
Two schemas of temporary constrain. (**a**) top view of temporary constraint 1, (**b**) top view of temporary constraint 2, (**c**) lateral view of temporary restraint, (**d**) schematic diagram of connection mode of "Beam MPC" in ABAQUS.

**Figure 20 materials-15-08265-f020:**
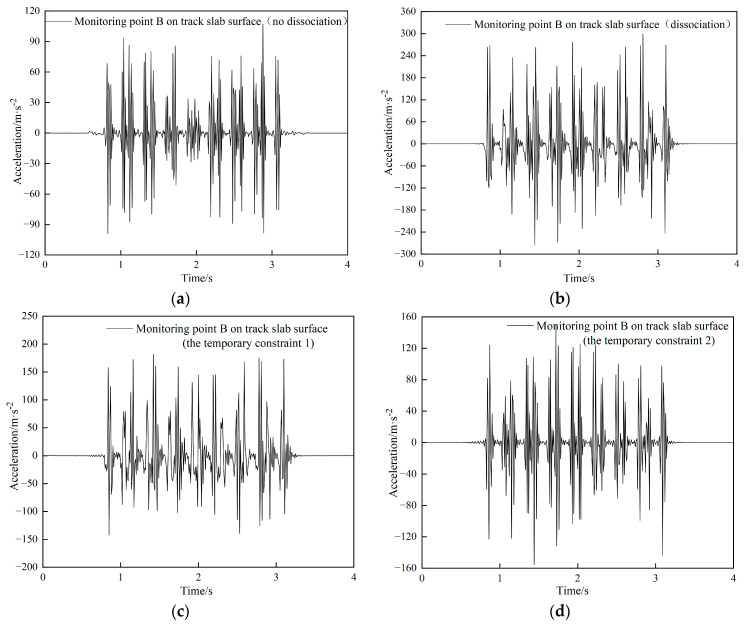
Acceleration time history curve of measuring point B under four different conditions: (**a**) no dissociation, (**b**) dissociation, (**c**) temporary restraint 1, (**d**) temporary restraint 2; Vertical displacement time history curve of measuring point B under four different conditions: (**e**) no dissociation, (**f**) dissociation, (**g**) temporary restraint 1, (**h**) temporary restraint 2.

**Figure 21 materials-15-08265-f021:**
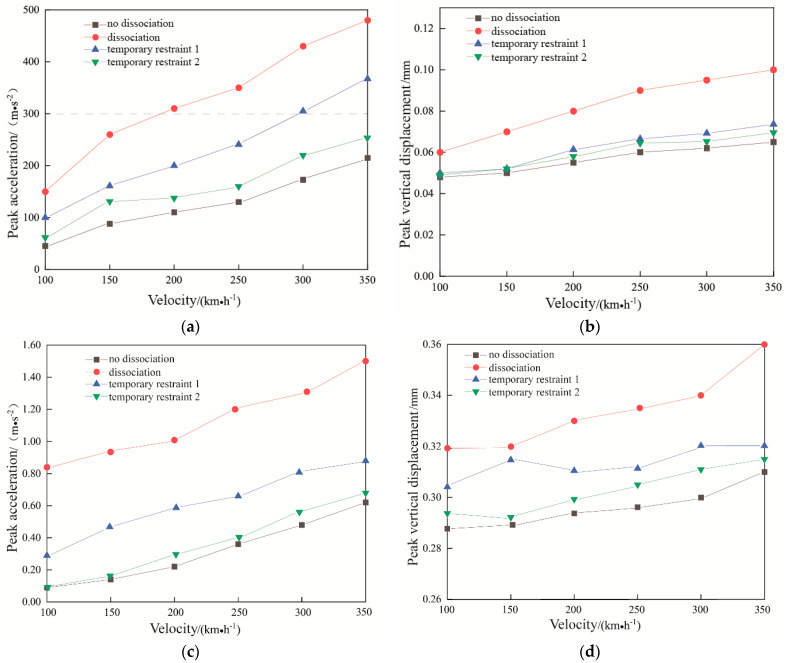
Peak acceleration and displacement curves at different train speeds. (**a**) the peak acceleration curve, and (**b**) peak displacement curve at the center of the track slab in the middle of the mode; (**c**) peak acceleration curve, and (**d**) peak displacement curve at the corresponding position of the subgrade bed.

**Table 1 materials-15-08265-t001:** Structural dimensions and material properties of track system.

Track Components	Size (mm)	Elastic Modulus (GPa)	Poisson’s Ratio	Density (kg/m^3^)
Track slab	6450 × 2550 × 200	36	0.2	2500
Mortar layer	2550 × 30	7	0.2	1950
Base plate	3250 × 300	22	0.2	2500
Rail bearing platform	500 × 250 × 60	36	0.2	2500
Wide joint	210 × 100	36	0.2	2500
Narrow joint	50 × 100	36	0.2	2500
Rebar	∅20	210	0.3	7800
Rail	CHN60	210	0.3	7830

**Table 2 materials-15-08265-t002:** Parameters of soil mass.

Name	Thickness/m	Weight/(kN/m3)	Modulus of Elasticity/GPa	Poisson’s Ratio	Cohesion/kPa	Angle of Internal Friction/(°)
Surface of subgrade bed	0.4	19.5	0.25	0.3	32	32
Bottom of subgrade bed	2.3	19.0	0.20	0.3	26	25
The body of the subgrade	1.8	18.5	0.15	0.28	25	23

**Table 3 materials-15-08265-t003:** CRH380 high-speed train parameters.

Parameters	Numeric Value	Parameters	Numeric Value
Marshalling type/carriage	8	Total length (m)	203.0
Bogie shafts (m)	2.5	Axle load (t)	≤15
Intermediate vehicle length (m)	25.0	Steering rack center distance (m)	17.5

## Data Availability

Not applicable.

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
