# Peer review of "Line Shape Analysis and Dynamic Response of Ballastless Track during Jacking Rectification Fixing"

_materials, 2022, doi:10.3390/ma15228265_

Round 1
Reviewer 1 Report
The manuscript is well described; however, the authors should highlight the purpose of the study, selection criteria, and novelty in the introduction. Figs. 1,2 can be modified with a more detailed explanation. A table or chart can be provided to compare the existing and proposed methods.
Reviewer 2 Report
In this paper, the authors have studied the railway line deformation and dynamic response of ballastless track structure under train load during jacking rectification fixing by using the 3D finite element modeling. Although the subject of this paper is interesting, the reviewer believes that this paper is not suitable for publication at the present format. The following comments and questions should be addressed before publication.
Comments:
1) The verification of the finite element model is not strong. It is not clear how much is the error of the presented finite element model in comparison to the experimental data. For the verification, the readers are just referred to ref. [30]. It is recommended that the authors report both experimental and numerical results in a table or chart and compare them together.
2) The details of finite element modeling should be cited in section 2. The constitutive laws used for modeling the behavior of constituent parts of the slab ballastless track and subgrade should be stated clearly.
3) Quality of Fig. 11, 12, 18 and 20 is not good. The numbers and texts are very small and cannot be read clearly.
4) No reference is cited for equations (3) and (4).
Reviewer 3 Report
The manuscript titled “Line Shape Analysis and Dynamic Response of Ballastless Track During Jacking Rectification Fixing” deals with the railway line deformation and dynamic response of ballastless track structure under train load during jacking rectification fixing. The manuscript is dealing with an extremely interesting topic and I believe it could be improved considering the following points:
The intro duction is well written but it is lacking in literature and reference of similar studies.
Numerical model – requires more details about the mesh size, material models of the different parts.
“The rail is in the form of CHN60 section [22], with a mass of 60.64 kg·m-1, a crosssectional area of 77.45 cm2 and a horizontal axis moment of inertia of 3217 cm4” what about the vertical axis? It will be important for the lateral bending
Rebar is mentioned in table 1 but what’s the use of it?
interface stiffness of 63.039 MPa/mm – units to be checked
“For dissociation, the contact between the base plate and the subgrade bed is described by classical friction theory-Coulomb friction theory” not clear.
“deviation of the corresponding track slab at the single point jacking position is calculated as shown in Fig. 4.” Not clear this section and the meaning of the curves in the graph.
The use of dissociation must be explained. Is it a technical term?
Figure 5 and 6 is showing displacement ibut not clear where these values are derived from
Is there any validation of the single jacking point?
Multi-point jacking should be in several location, but the deflection does not seem to show this.
Fig 11 not clear as each detail “is not providing the required information to understand what they are
Is the interface damage algorithm validated?
“In order to study the value of the jacking force at the bonding condition of the mortar layer and the wide-narrow joint interface in the single point jacking process, a series of working conditions are simulated.” Are these conditions validated wit experiments?
“Damage analysis for wide-narrow joint during multi-point jacking” but the details on the contact and the damage are not given
“The continuous axle weight load generated by the train is shown in Equation 3:” you might need to check the English
. The calculated results are in general agreement with the measured vibration acceleration time course curves in the literature [30] – validation must be provided with more details
Modelling strategy throughout the manuscript not well presented.
What is the step adopted to model the moving load? Is it explicit or implicit? This is not clear
Fig. 20. The time-history curve of dynamic response of track slab – is this the output of your model?
Conclusions might be shortened
Round 2
Reviewer 2 Report
The authors have considered all reviewer's comments in the preparation of the revised version of paper. It is recommended for publication at the present format.
Reviewer 3 Report
thanks for considering my comments